# Evolution of Bacterial Persistence to Antibiotics during a 50,000-Generation Experiment in an Antibiotic-Free Environment

**DOI:** 10.3390/antibiotics11040451

**Published:** 2022-03-27

**Authors:** Hugo Mathé-Hubert, Rafika Amia, Mikaël Martin, Joël Gaffé, Dominique Schneider

**Affiliations:** Université Grenoble Alpes, CNRS, UMR 5525, VetAgro Sup, Grenoble INP, TIMC, 38000 Grenoble, France; amiarafika@gmail.com (R.A.); mikael.malek@univ-grenoble-alpes.fr (M.M.); joel.gaffe@univ-grenoble-alpes.fr (J.G.); dominique.schneider@univ-grenoble-alpes.fr (D.S.)

**Keywords:** antibiotic persistence, evolution, *Escherichia coli*, beta-lactam, ampicillin, fluoroquinolones, ciprofloxacine, Start Growth Time, bacterial quantification

## Abstract

Failure of antibiotic therapies causes > 700,000 deaths yearly and involves both bacterial resistance and persistence. Persistence results in the relapse of infections by producing a tiny fraction of pathogen survivors that stay dormant during antibiotic exposure. From an evolutionary perspective, persistence is either a ‘bet-hedging strategy’ that helps to cope with stochastically changing environments or an unavoidable minimal rate of ‘cellular errors’ that lock the cells in a low activity state. Here, we analyzed the evolution of persistence over 50,000 bacterial generations in a stable environment by improving a published method that estimates the number of persister cells based on the growth of the reviving population. Our results challenged our understanding of the factors underlying persistence evolution. In one case, we observed a substantial decrease in persistence proportion, suggesting that the naturally observed persistence level is not an unavoidable minimal rate of ‘cellular errors’. However, although there was no obvious environmental stochasticity, in 11 of the 12 investigated populations, the persistence level was maintained during 50,000 bacterial generations.

## 1. Introduction

The current evolution of bacterial resistance to antibiotics brings humanity back to a situation reminiscent of the ‘pre-antibiotic era’ [1,2], mostly owing to the impressive bacterial adaptive properties. This alarming situation results in the death of more than 700,000 people every year. A worst-case scenario anticipates that this number might rise up to 10 million deaths per year by 2050. In Europe, in 2015, this caused ~30,000 deaths per year, twice that in 2007 [3].

To cope with high antibiotic concentrations, bacteria rely on various processes that lead to different population dynamics [4,5,6]. These differences can be used to tentatively classify them. (I) Resistance allows bacteria to grow under such antibiotic stress. (II) Tolerance slows down the death rate of most of the population. (III) Persistence involves only a tiny proportion of the population, from 10^−8^ to 10^−2^ in natural populations [7], that benefits from a low death rate associated with low metabolic activity. This persistent state allows bacterial cells to cope with a broad range of stresses, albeit being genetically susceptible. Indeed, the size of a population exposed to antibiotics and containing persister cells first quickly decreases, owing to the death of susceptible and metabolically active cells, before experiencing a much slower decrease due to the death of persister quiescent cells. This physiological state is not heritable; hence, if a population regenerated from the persister sub-population is re-exposed to the antibiotic, a similar phenotypic heterogeneity will typically be observed again. This phenomenon allows growth rescue and restart after many stresses, including nutrient depletion, temperature change, acid, oxidative or osmotic challenges, phage infection, and exposure to heavy metals or antibiotics [6,8].

Persistence is a natural process that evolved before the anthropogenic use and abuse of antibiotics. It might exist in all living organisms as it has been observed in all studied bacterial species [6,9], in eukaryotic cells resulting in cancer relapse [9,10,11,12], and even accidentally in digital organisms that learned to play dumb randomly at a low rate [13]. Understanding the evolutionary forces driving persistence emergence and maintenance may have far-reaching sanitary consequences. Indeed, cellular persistence contributes to the relapse of many infections [14,15,16,17], including recurrence of mycobacterial infections in 10% of patients [18]. Cellular persistence is also involved in the persistence of some infections, although these can also rely on tolerance and resistance; hence, cellular persistence and persistent infections have to be distinguished [14]. Moreover, persistence may have indirect effects on the evolution of antibiotic resistance as it can increase: (i) mutation rates, (ii) horizontal gene transfer rates, and (iii) survival to antibiotics of susceptible cells by allowing repeated antibiotic exposures that select for progressive increase of antibiotic resistance [6,19]. Therefore, persistence is likely an important driver of antibiotic resistance evolution.

Although first described more than 70 years ago [20], the evolutionary forces driving persistence are still not well understood. It is important to emphasize that persistence is a typical case of phenotypic stochasticity, i.e., genetically identical bacterial cells that behave differently in given environments. This phenotypic stochasticity may either pre-exist to the stress (type I persister cells) or be induced by it (type II persister cells) [15,21]. Beyond persistence, stress-induced phenotypic stochasticity is a general feature of living organisms that is observed in many taxa and organization levels, such as increased developmental noise in plants [22], bears [23], fishes [24], and Drosophila [25] (for reviews, see [26,27,28,29,30]).

There is a current debate about the evolutionary meaning of phenotypic stochasticity [31,32,33,34], including persistence. It is hypothesized to be either an unavoidable consequence of biological constraints or an adaptive process related to bet-hedging. According to the first hypothesis, any phenotype has a minimal amount of random variance. In addition, this variance can be increased by stresses which may lead to random abnormal phenotypes by causing cell alterations. Indeed, stresses will hamper mechanisms selected to reduce phenotypic stochasticity (developmental stability [35]), thereby resulting in a minimal amount of phenotypic stochasticity. For example, acid and temperature stresses can lower protein conformational stability [36]. In the second hypothesis, by increasing the probability that few individual cells are adapted to the stress, this variability may be advantageous and thus selected for, leading to the evolution of bet-hedging [37]. For example, cells from all life kingdoms have a mistranslation mechanism that is activated as a stress response [30]. This discrepancy between the two hypotheses is particularly relevant for persistence [7]. Levin et al. [38] emphasized that persistence might be related to errors in cellular processes that block cells in almost inactive states. In agreement, a large number of mutations can result in increased persistence frequency [39], while no mutation has ever been found to prevent it [40], supporting the hypothesis that persistence is merely a result of biological constraints. On the other hand, some persister cells can actively pump antibiotics out from cells [41,42], and their proportion in the population may evolve rapidly, suggesting an adaptive process. Indeed, chronic bacterial infections evolved an increased rate of persistence after antibiotic treatment [43,44,45,46].

Laboratory evolution experiments confirmed the high persistence evolvability. In the presence of antibiotics, a 100- to 1000-fold increase in persistence has indeed been reported after only two to three cycles of antibiotic selection (less than five days) [47]. In contrast to antibiotic resistance, the production of a small proportion of persister cells entails low cost to the population. Therefore, restoring the initial level of persistence is a slow process, requiring hundreds to thousands of generations [47]. Hence, long-term experimental evolution in the absence of killing stressors, such as antibiotics, is particularly relevant to study persistence evolution. Here, we used the long-term evolution experiment (LTEE) that was initiated in 1988 by Richard Lenski, using the two *Escherichia coli* clones REL606 (Ara−) or REL607, the latter being a spontaneous Ara+ revertant of REL606. From each of these two clones, six populations were initiated and are propagated in a glucose-limited, antibiotic-free environment, by serial daily transfer in the same fresh medium [48]. According to their respective ancestor, these twelve populations are named Ara−1 to Ara−6 and Ara+1 to Ara+6. We recently showed that susceptibility to many antibiotics increased over time in these conditions, in which bacteria were selected for faster growth for more than three decades [49].

We investigated the evolution of persistence for ampicillin and ciprofloxacin by comparing their effects in the ancestor and evolved clones sampled in each of the twelve populations up to 50,000 generations. We improved a high-throughput methodology [50] to estimate the prevalence of persisters while accounting for growth rate heterogeneity.

## 2. Results

We performed two sets of analyses. First, in the ‘LTEE-50K’ analysis, we compared persistence to the two antibiotics ampicillin and ciprofloxacin of one clone, sampled at 50,000 generations from each of the 12 populations of the LTEE to their respective ancestors, REL606 and REL607. Second, in the ‘Ara−2_S_L’ analysis, we investigated the persistence level within the population called Ara−2 (Table 1), in which an adaptive diversification event occurred. Indeed, two phenotypically distinct ecotypes, called S and L, emerged by generation 6500 and have co-existed ever since [51]. We sampled 10 evolved clones from Ara−2, including one from each generation 2000 and 5000 before the emergence of this polymorphism, and one S and L clone from each generation 6500, 11,000, 20,000 and 50,000 (Table 1).

To quantify persistence after a 5-hour antibiotic exposure, we performed a ten-fold dilution cascade, from 10^0^ to 10^7^. For each dilution, we recorded the bacterial culture revival, i.e., the presence/absence of growth after 24 h and, if revived, we quantified the number of persister cells by analyzing the growth curve ($\hat{\mathrm{initial}\;\mathrm{OD}}$ approach described hereafter). Absence of growth after 24 h was related to either the absence of persister cells or a number of persister cells too low for their revival and detection. In this case, we set the number of persister cells to zero.

### 2.1. Validation of the $\hat{\mathrm{initial}\;\mathrm{OD}}$ Approach to Quantify Persister Cells

We adapted the approach used by Hazan et al. [50] by recording the intercept of a linear model fitted to the $log_{2}$ of the growth curve, which is an estimate of the initial OD (hereafter, $\hat{\mathrm{initial}\;\mathrm{OD}})$, instead of recording the time needed to reach a threshold OD, called the Start Growth Time (SGT). This approach avoids biases induced by growth rate variations and can detect tiny variations in $\hat{\mathrm{initial}\;\mathrm{OD}}$ (Appendix B). However, as the SGT approach [50], it assumes that the lag time for cell regrowth is unaffected by the antibiotic treatment, although it is known to differ between persister and non-persister cells [21]. We accounted for this approximation by referring, for this estimated number of persister cells (CFU number), to an Equivalent Number of Normal Cells (#EqNC), i.e., the initial number of cells that would have yielded the same $\hat{\mathrm{initial}\;\mathrm{OD}\;}$ in the absence of antibiotic exposure.

To validate this $\hat{\mathrm{initial}\;\mathrm{OD}}$ approach, we checked that, on average, the ten-fold dilution series yielded ten-fold differences in the #EqNC. In that case, the average of the slopes of models predicting $\log_{10}\left(\#\mathrm{EqNC}\right)$ by $\log_{10}\left(\mathrm{dilution}\right)$ should be equal to one. We found an average of 0.97, with a bootstrap of 95% CI of [0.90; 1.05].

### 2.2. Evolution of Persistence in the ‘LTEE-50K’ Analysis

For each growth curve, we estimated $\hat{\mathrm{initial}\;\mathrm{OD}}$ and used standard curves specific to each strain for conversion into CFUs (#EqNC, Appendix B). We used a linear mixed model to estimate the mean #EqNC and its confidence interval for each clone (Table 1) and treatment (ampicillin, ciprofloxacin, no antibiotics). This model predicts the $\log_{2}\left(\#\mathrm{EqNC}\right)$ of each growth curve as a function of, fixed effects, $\log_{10}$ of the dilution, antibiotic treatment, clone ID, and two second-order interactions with the antibiotic treatment, and random effects on these fixed effects. All fixed effects were highly significant (Table 2).

#### 2.2.1. Overall Trends in the Persistence Level to Ampicillin and Ciprofloxacin

We analyzed the estimated #EqNC for each investigated clone and treatment (coefficients of the model summarized in Table 2 and shown Figure 1). We found that the overall level of persistence to ciprofloxacin and ampicillin was similar (bootstrapped paired *t*-test implemented in the R package, MKinfer: *p*-value = 0.22) and positively correlated (Spearman rank correlation = 0.508; *p*-value = 0.014; Figure 1). However, this pattern varied according to the clones. Six of the twelve populations evolved a mutator phenotype owing to mutations in DNA repair genes before 50,000 generations (Table 2; [52]). We found no association between the persistence level and the mutator/non-mutator state of the populations (linear mixed model; *p*-value of the interaction between mutator state and treatment = 0.94). Nevertheless, the positive correlation between persistence to ciprofloxacin and ampicillin was mostly driven by the non-mutator clones (Spearman rank correlations = 0.87 and 0.08; *p*-values = 0.003 and 0.78, respectively, for the non-mutator and mutator clones). A permutation test comparing these two correlations yielded a *p*-value of 0.037 (50,000 permutations of $\;\left|\sigma_{mutator}-\sigma_{non-mutator}\right|$).

The among-strain variability for persistence to ciprofloxacin was higher than that to ampicillin (Ansari–Bradley test AB = 331; *p*-value = 0.016; Figure 1). There was no correlation between persistence and resistance to these antibiotics [49] (Spearman rank correlations = 0.365 and 0.074; *p*-values = 0.22 and 0.81 for ampicillin and ciprofloxacin, respectively). 

For each clone (Table 2), the abundance of persister cells to ampicillin and ciprofloxacine was quantified by the ratio between the #EqNC in the treatment and the control. Each dot corresponds to a given clone and gives the abundance of persister cells to the two antibiotics. Dotted lines give the 95% CI. These values were obtained from the coefficients of the models (Table 2).

#### 2.2.2. Evolution of Persistence after 50,000 Generations of Evolution

We first showed that there was no significant difference in the level of persistence between the two ancestral clones REL606 and REL607 (*p*-values = 0.20 and 1 for ampicillin and ciprofloxacin, respectively). Then, we performed both comparisons of each evolved clone to each of the two ancestral strains and pairwise comparisons among the 13 clones sampled at generation 50,000 using the model coefficients (Table 2; Figure 2). The evolution of persistence to ciprofloxacin of the S clone from population Ara−2 was the only significant difference compared to the ancestors (*p*-value = 0.01; Figure 2). However, the pairwise comparisons among clones from generation 50,000 had more statistical power and detected significant changes in the level of persistence to ciprofloxacin. Persistence was: (i) higher in the Ara+4 clone than in clones from populations Ara−3, Ara−4, Ara−5, Ara−6, Ara+2, Ara+3, and the S clone from Ara−2; (ii) higher in the Ara+5 and Ara+6 clones than in clones from populations Ara−4 and Ara+2, and the S clone from Ara−2 (Appendix A).

### 2.3. Evolution of Persistence in the ‘Ara−2_S_L’ Analysis

We investigated the interplay between the adaptive diversification event that occurred in population Ara−2 and the evolution of persistence. The emergence of diversification was detected between generations 5000 and 6500, leading to the co-existence of two ecotypes, called Ara−2L and Ara−2S [51]. From generation 11,000 to 50,000, each Ara−2S-sampled clone had a significantly lower level of persistence to ciprofloxacin compared to both the ancestors and the corresponding contemporary Ara−2L sampled clone, while there were no significant differences for persistence to ampicillin (Figure 3).

## 3. Discussion

We investigated the evolution of bacterial persistence during the LTEE over 50,000 generations, corresponding to 22 years during which bacterial cells were maintained in a defined antibiotic-free environment [53]. Environmental variations included the daily cycles of feast and famine and changes produced by bacteria themselves, as for example, secretion of metabolic byproducts [54]. We improved a high-throughput method to quantify persister cells by accounting for growth rate heterogeneity. Despite evolutionary potential for lower persistence, it was not observed in most cases in this antibiotic-free environment.

Studying persistence is complex because (i) persister cells are genetically identical to non-persistent cells; (ii) persister cells are rare in entire populations; (iii) the persistence state is not heritable; and (iv) once in the persistence state, cells do not multiply. Here, we improved a previously published method [50] to quantify persister cells. This approach relied on analyzing the growth curve of the bacterial population that recovered from the stress that killed non-persistent cells. The more persister cells, the faster the growth of the reviving population, which thereby becomes detectable. This approach assumed that the growth rate is constant among treatments. We, however, showed that this was not the case for persister cells (Appendix A). Therefore, we fitted linear models on the log_2_ of each growth curve, which estimates the growth rate and initial population density.

We detected no relationship between the levels of persistence and antibiotic resistance [49], thereby showing that the two phenomena are related to two different pathways in the conditions of the LTEE, where there is no antibiotic pressure but selection for growth. Similar results were observed in natural strains of *Pseudomonas* spp. [55]. 

Overall, among the LTEE clones, the level of persistence to the two antibiotics was similar and positively correlated, suggesting similar physiological pathways. However, the positive correlation was only driven by non-mutator strains, and the among-strain variation was significantly higher for ciprofloxacin than ampicillin. In addition, we detected no significant differences among clones for ampicillin, by contrast to ciprofloxacin. This suggests the existence of different types of persister cells, some being shared between ampicillin and ciprofloxacin, and some that are specific to each antibiotic. The observed among-clone variation would result from both types of persister cells: those common to the two antibiotics explaining the correlation in the non-mutators and those specific to each antibiotic explaining the difference between the two antibiotics in among-clone variance. In agreement with this diversity of persister cells, it has been observed that a large panel of pathways can trigger persistence [15], and the correlation between persistence to different antibiotics is variable. Hence, a marginally significant correlation between the persistence to ampicillin and nalidixic acid was found in environmental samples of *E. coli* [56], but no correlation between the persistence to these two antibiotics and to ciprofloxacin, albeit ciprofloxacin and nalidixic acid share a similar mechanism of action. A similar analysis by Stewart and Rozen [57] revealed no correlation in the level of persistence to ampicillin, streptomycin, and norfloxacin.

Persistence in the ancestral strains REL606 and REL607 may originate from either past selection by environmental stochasticity [43,44,45,46,47] or results from biological constraints that leads to a minimal amount of unavoidable ‘cellular errors’ [38,39]. In the former case, we would expect no evolution of persistence and a higher amount of persister cells in mutator clones, because the mutational target for higher persistence is large [39]. In addition, if persistence in the ancestor results from adaptation to former environmental stochasticity, the naïve prediction would be a reduced persistence level in most, if not all, LTEE populations that evolved under a strong selection for improved growth during 22 years in a constant environment. Indeed, bacteria have a strong capacity to adapt to cyclic or correlated environmental changes [58,59,60,61], and shift to a random dormancy state should be costly as any growing mutant among other dormant cells would have a higher growth rate.

By contrast to these two alternative predictions, we detected no relationship between the persistence and mutator state, and only one of the 13 analyzed clones (from the Ara−2S ecotype) showed significant evolution toward lower persistence. Evolved clones from the two populations, Ara−4 and Ara+2, evolved lower persistence that was only significant when compared to clones that evolved higher persistence (from populations Ara+4, Ara+5, and Ara+6). These results show an evolutionary potential for both higher and lower persistence, but in most cases, the stable LTEE environment did not select for low persistence.

The only clone revealing lower persistence is the 50,000-generation Ara−2S clone. Interestingly, its relative fitness compared to its contemporary clone from the Ara−2L ecotype is higher during the stationary phase than the exponential phase [62,63]. Indeed, while the Ara−2L clone was starving from glucose, the Ara−2S clone consumed the acetate produced during growth on glucose [64]. Hence, it might favor dormancy of the Ara−2L clone during the stationary phase to lower both its energy consumption and death rate, while the Ara−2S clone was actively growing on acetate. This hypothesis is particularly appealing since starvation has been shown to be a main natural cause of persistence evolution [6,21,65]. Hence, persistence may provide benefits to starvation in natural environments because feast and famine phases are poorly predictable. By contrast in the LTEE, bacteria experience a seasonal and predictable environment every day since more than three decades, oscillating between exponential phase (feast) and stationary phase (famine). Because of such reduced stochasticity in the LTEE between the different growth seasons, randomly switching a small proportion of cells into dormancy may not be a beneficial strategy.

Further studies may investigate the influence of the daily starvation phase in the maintenance of persistence during the LTEE. Alternatively, the observed variability among the evolutionary trajectories of persistence might result from pleiotropic interactions between persistence and other selected traits. According to this scenario, the variability in evolutionary trajectories of different populations would result from some contingencies and would be a side effect of the divergence of populations. Understanding these pleiotropic interactions would be very useful to optimize the treatments of infections or cancers [66]. Indeed, this might allow the pre-evolving of an infecting bacterial cell population or a cancer cell population, to have a low persistence to the future treatment or to be more sensitive to molecules such as mannitol, that start being used against persistent cells [66,67].

The conundrum of the unexpected maintenance of persistence in a stable environment over 50,000 bacterial generations, albeit evolutionary potential for decreasing the amount of persistence as observed in the Ara−2S clone, highlights the LTEE importance for challenging, testing, and improving our understanding of Darwinian evolution.

## 4. Materials and Methods

### 4.1. Strains

We used *E. coli* clones that are derived from the LTEE initiated by Richard Lenski. During that experiment, each of the two ancestral clones, REL606 (Ara−) and REL607, the latter being a spontaneous Ara+ revertant of REL606 [53,68], was used to initiate six populations. Since 1988, these 12 populations are propagated in 10 ml of DM25 medium (Davis Minimal broth supplemented with glucose at 25 mg/L) by a 1/100 daily serial transfer in the same fresh media, thereby yielding $\log_{2}\left(100\right)\approx 6.64$ generations every day.

The 23 clones we used in this study are the two ancestral clones REL606 and REL607, one evolved clone sampled at 50,000 generations from 11 of the 12 LTEE populations, and 10 evolved clones from the population called Ara−2 (Table 1). Indeed, the population Ara−2 experienced an adaptive diversification event, during which two phenotypically distinct ecotypes, called S and L, emerged by generation 6500 and have co-existed ever since [51]. Therefore, we sampled one Ara−2 evolved clone from generations 2000 and 5000 before the emergence of this polymorphism, and one S and L clone at each of the generations, 6500, 11,000, 20,000, and 50,000 (Table 1).

### 4.2. Measuring the Proportion of Persister Cells

For each LTEE-derived clone, we estimated the amount of persister cells after exposure to each of the two bactericidal antibiotics belonging to two different families, ampicillin and ciprofloxacin, by hypothesizing that the larger the persister population is, the faster regrowth will happen and therefore be detectable by optical density [49] (see below).

All clones were grown in DM1000 medium (Davis Minimal broth supplemented with glucose at 1 g/L) at 37 °C and 180 rpm. This high glucose concentration improves the accuracy of the OD measure during regrowth after antibiotic exposure. After overnight culture, stationary-phase cells were diluted at 1/1000 in 500 mL DM1000. After 2.5 h of growth (mid-exponential phase), we divided each culture into nine 5 mL tubes that were subdivided into three groups, i.e., three technical replicates per group. In the two first groups, we added either ampicillin or ciprofloxacin at a final concentration of 100 and 1 µg/mL, respectively. The third group was used as an antibiotic-free control. After 5 h at 37 °C and 180 rpm, we removed antibiotics by three successive washes consisting in 5-minute centrifugations at 1500 g, with a removal of 90% of the supernatant and resuspension of the pellet in antibiotic-free DM1000. Finally, cells were resuspended in 200 µL DM1000, and each tube content was transferred into a well of a 96-well microplate (Thermo Scientific, Waltham, MA, USA, 260860). We performed for each initial tube a ten-fold dilution cascade from 10^0^-to-10^7^-fold, and monitored growth in an Infinite M200 microplate reader (Tecan^®^, Mennedorf, Switzerland) to quantify the proportion of persister cells for each clone. We recorded the OD_600_ every 15 min for 24 h which we used as the ‘time’ variable in our analysis (see below). In addition, for each antibiotic-free control tube, we estimated the CFU number to compute the relationship between the number of cells and the time after which regrowth was detected by OD. We performed at least three biological replicates for each clone.

### 4.3. Rationale of Data Analyses

All analyses were performed to: (i) compare the persistence frequency in each of the 12 evolved clones sampled at 50,000 generations to the corresponding ancestor REL606 or REL607, and (ii) analyze the evolutionary dynamics of persistence in the population Ara−2, in which the S and L ecotypes emerged by generation 6500 and co-exist since then [69]. We refer, hereafter, to these two analyses as LTEE-50K and Ara−2_S_L, respectively (Table 1).

### 4.4. Quantification of the Population Size of Persister Cells

We improved a previous approach [50] that is similar to qPCR as it relies on recording the time needed to detect an increase in the signal (here, OD_600_), which is proportional to the initial amount of material (here, the number of persister cells). Hence, the time needed to reach a given OD_600_ threshold is defined as the Start Growth Time, SGT [50]. This approach, however, assumed that persister cells have both a growth rate and a lag phase that are similar to the cells of the cultures used for the standard curve, albeit they were not exposed to antibiotics (Figure A1e). Comparing the growth rates of each strain in each treatment showed that persister cells actually had a slower growth rate than other cells. For unknown reasons, this differential effect was stronger for ciprofloxacin than ampicillin (Appendix A). Therefore, we developed an alternative approach based on a statistical model that predicted the $\log_{2}$ of the OD_600_ observed during exponential growth as a function of time (Appendix A; heuristic selection of the exponential growth phase). These models estimate both the growth rate (slope) and initial OD_600_ (intercept; hereafter, $\hat{\mathrm{initial}\;\mathrm{OD}}$) for each growth curve. This approach can detect tiny variations in initial OD as they increased during exponential growth (for more details, see Appendix B). Using this approach, we obtained an $\hat{\mathrm{initial}\;\mathrm{OD}}$ for each growth curve and converted it into cell numbers using standard curves obtained for each strain by quantifying both $\hat{\mathrm{initial}\;\mathrm{OD}}$ and CFUs in dilution series (Figure A1d).

However, while this approach accounted for growth rate heterogeneity, it still assumed that the mean lag time for cell regrowth was identical in all strains and treatments. Hence, we refer to this estimated number of persister cells (number of CFU) as an Equivalent Number of Normal Cells (#EqNC), corresponding to the initial number of cells that, in the absence of antibiotics, would have produced the same $\hat{\mathrm{initial}\;\mathrm{OD}}$. In the absence of growth after 24 h, the #EqNC was set to zero. To validate this $\hat{\mathrm{initial}\;\mathrm{OD}}$ approach, we checked that the ten-fold dilution series indeed yielded ten-fold differences in the estimated amount of persister cells (#EqNC). Specifically, we used the $\hat{\mathrm{initial}\;\mathrm{OD}}$ approach to estimate #EqNC separately for each growth curve. Then, for each dilution series, we fitted a model predicting $\log_{2}\left(\#\mathrm{EqNC}\right)$ by $\log_{10}\left(\mathrm{dilution}\right)$. If the ten-fold dilutions resulted in an averaged ten-fold difference in the number of persister cells, these models should have a slope of one that we checked by bootstrapping the slopes.

### 4.5. Estimating and Comparing Persistence

We estimated the level of persistence to ampicillin and ciprofloxacin in each strain by fitting a linear mixed model predicting the $\log_{2}\left(\#\mathrm{EqNC}\right)$ which had, (i) as fixed effects, the log_10_ of the dilution factor, the antibiotic treatment, the clone, and the two second-order interactions with antibiotic treatments, and (ii) as random effects, the ‘replicates’ on the intercept, the three fixed effects of ‘dilution’, ‘treatment’, and their interaction. The intercept of these models estimates the mean $\log_{2}\left(\#\mathrm{EqNC}\right)$ in the non-diluted sample, i.e., when the log_10_ of the dilution is equal to zero.

We fitted this model with the lme4 package [70] of the R version 4.0.3 and tested the significance of effects with F tests and the Satterthwaite approximation for degrees of freedom [71,72] (Table 2). We compared clones to each other using the R package ‘multcomp’ (version 1.4-17; [73]). For each antibiotic, we compared each evolved strain to the two ancestors, REL606 and REL607. This first set of tests was used to, (i), check for the absence of significant differences between the two ancestors, and, (ii), assess whether the evolved clones were significantly different from their ancestors. To be conservative when detecting changes, only clones that were significantly different from both ancestors were considered as being significantly different. We also performed pairwise comparisons between 50,000-generation clones to detect groups of clones that evolved in opposite directions. Finally, we used this test to compare contemporary co-existing clones, Ara−2 L and S.

### 4.6. Relationship between Persistence and Mutator Phenotype

To test for an effect of the mutator phenotype on persistence, we re-fitted the linear mixed model predicting $\log_{2}\left(\#\mathrm{EqNC}\right)$ by setting the two variables, (i) clone and (ii) interaction between clone and treatment, as a random instead of fixed effect, and by adding, as a fixed effect, the mutator state and its interaction with the treatment. Furthermore, we assessed the effect of the mutator state on the correlation between persistence to ampicillin and ciprofloxacin using Spearman rank correlations. Specifically, we measured and tested correlation (σ) among all clones and then separately among mutator and non-mutator clones. We tested for the significance of the difference between mutator and non-mutator clones using 50,000 permutations of the statistic $\;\left|\sigma_{mutator}-\sigma_{non-mutator}\right|$.

## Figures and Tables

**Figure 1 antibiotics-11-00451-f001:**
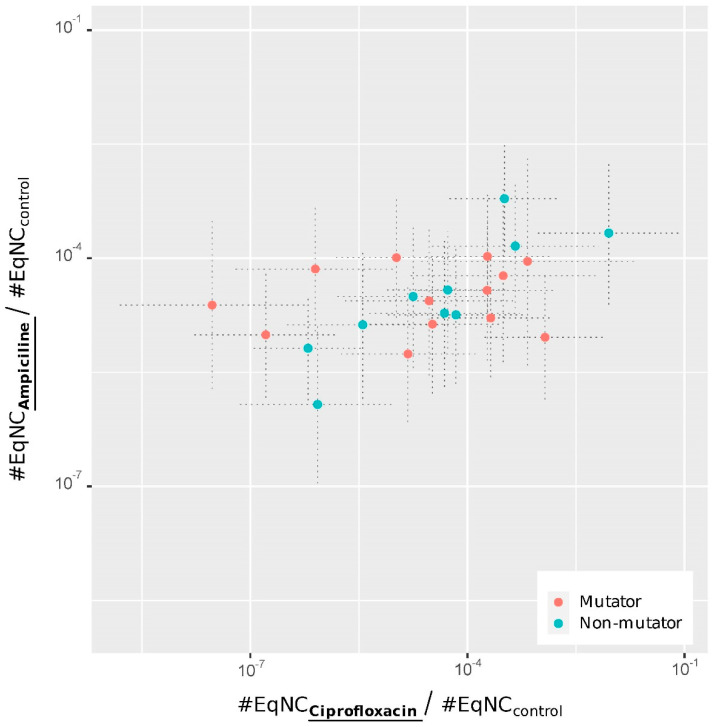
Persistence to ampicillin vs. ciprofloxacin in evolved clones sampled from each of the 12 LTEE populations.

**Figure 2 antibiotics-11-00451-f002:**
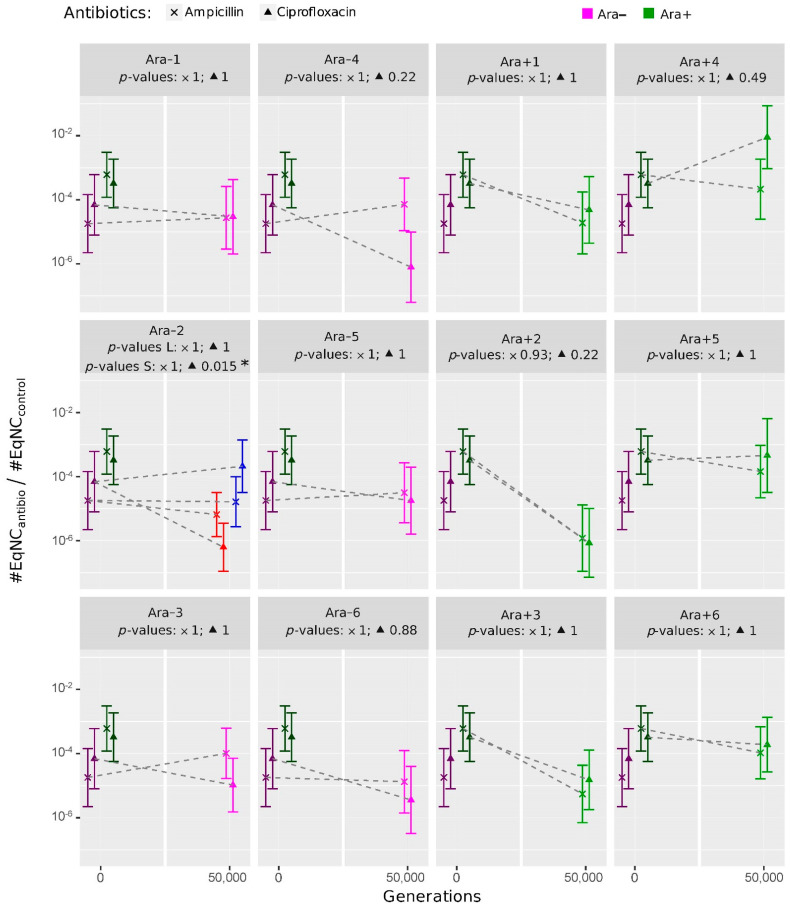
Evolution of persistence to ampicillin (×) and ciprofloxacin (▲) in evolved clones sampled from the 12 LTEE populations. For each antibiotic, we compared the level of persistence of each evolved clone sampled at generation 50,000 to the one in each of the two ancestors, REL606 and REL607, and to be conservative, only the least significant of the two comparisons was kept for each evolved clone. The *p*-values for each antibiotic are shown below the name of the population (and for each of the S and L ecotypes in population Ara−2, in red and blue, respectively). These values were obtained from the coefficients of the models summarized in Table 2. The 95% confidence intervals are shown. Dark symbols represent the ancestor strains, pink represent the clones from the Ara−1 to Ara−6 populations, and green represent the clones from the Ara+1 to Ara+6 populations. Significance codes: 0 ‘***’ 0.001 ‘**’ 0.01 ‘*’ 0.05 ‘.’ 0.1 ‘ ’ 1.

**Figure 3 antibiotics-11-00451-f003:**
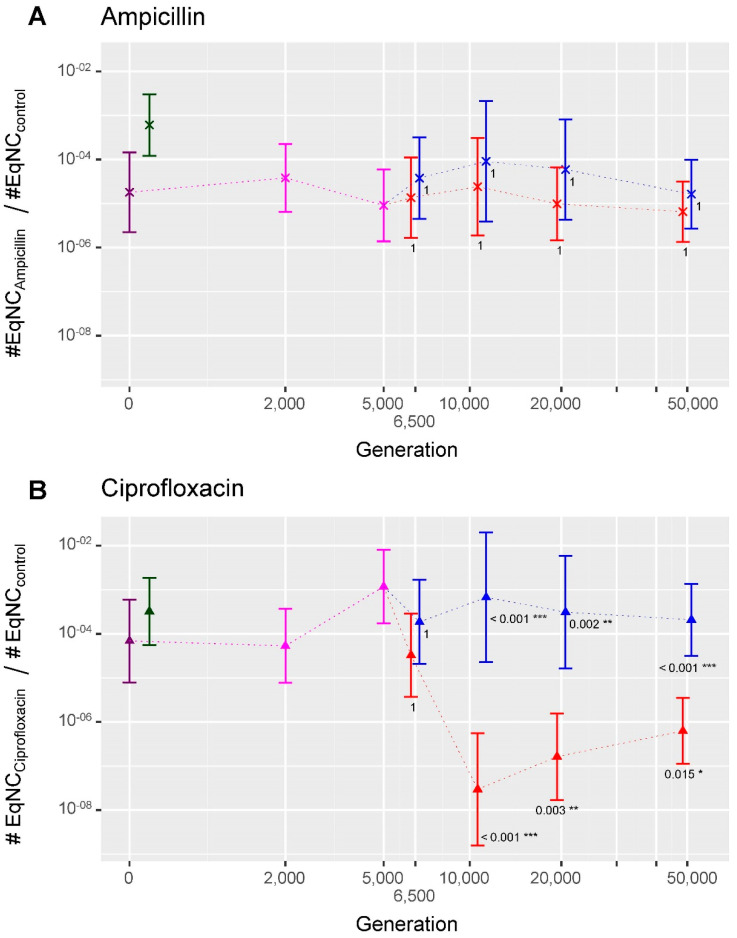
Evolution of persistence to ampicillin (**A**) and ciprofloxacin (**B**) in evolved clones sampled from the Ara−2 population. Dark symbols represent the ancestor strains REL606 and REL607; pink represent the Ara−2 evolved clones sampled before the adaptive diversification event; red and blue represent the evolved clones from the S and L ecotypes, respectively. *p*-values close to the Ara−2S evolved clones refer to the comparisons to the ancestors, and *p*-values close to the Ara−2L evolved clones refer to the comparison between the co-existing contemporary S and L evolved clones. The 95% confidence intervals are shown. Significance codes: 0 ‘***’ 0.001 ‘**’ 0.01 ‘*’ 0.05 ‘.’ 0.1 ‘ ’ 1.

**Table 1 antibiotics-11-00451-t001:** List of the LTEE-derived clones used in this study.

Clone	LTEE Population	Generation	Mutator State *	Analyses **	
REL606	Ancestor (Ara−)	0	N	LTEE-50K	Ara−2_S_L
REL607	Ancestor (Ara+)	0	N	LTEE-50K	Ara−2_S_L
11330	Ara−1	50,000	M	LTEE-50K	
1165A	Ara−2 (BC ***)	2000	N		Ara−2_S_L
2180A	Ara−2 (BC ***)	5000	M		Ara−2_S_L
6.5KS1	Ara−2 (S)	6500	M		Ara−2_S_L
6.5KL4	Ara−2 (L)	6500	M		Ara−2_S_L
11KS1	Ara−2 (S)	11,000	M		Ara−2_S_L
11KL1	Ara−2 (L)	11,000	M		Ara−2_S_L
20KS1	Ara−2 (S)	20,000	M		Ara−2_S_L
20KL1	Ara−2 (L)	20,000	M		Ara−2_S_L
13335	Ara−2 (S)	50,000	N	LTEE-50K	Ara−2_S_L
11333	Ara−2 (L)	50,000	M	LTEE-50K	Ara−2_S_L
11364	Ara−3	50,000	M	LTEE-50K	
11336	Ara−4	50,000	M	LTEE-50K	
11339	Ara−5	50,000	N	LTEE-50K	
11389	Ara−6	50,000	N	LTEE-50K	
11392	Ara+1	50,000	N	LTEE-50K	
11342	Ara+2	50,000	N	LTEE-50K	
11345	Ara+3	50,000	M	LTEE-50K	
11348	Ara+4	50,000	N	LTEE-50K	
11367	Ara+5	50,000	N	LTEE-50K	
11370	Ara+6	50,000	M	LTEE-50K	

* The mutator (M) or non-mutator (N) state is indicated. ** See text below: section “Rationale of data analyses”. *** BC, before co-existence.

**Table 2 antibiotics-11-00451-t002:** Tests of the fixed effects of the model analyzing the #EqNC.

Variable	df	*F*-Value	*p*-Value
$\log_{10}$(dilution)	1, 100.58	595.11	<0.001
Clone ID	23, 59.92	48.86	<0.001
Antibiotic	2, 1342.65	144.94	<0.001
$\mathrm{Antibiotic}\;\times \;\log_{10}$(dilution)	2, 131.47	7.63	<0.001
Antibiotic × clone ID	44, 399.89	9.45	<0.001

*p*-values of fixed effects are based on F-tests with Satterthwaite’s approximation. The corresponding numerator and denominator degrees of freedom (df) and statistics of the tests (*F*-values) are given.

## Data Availability

The data presented in this study have been made openly available in FigShare, at doi:10.6084/m9.figshare.19209690.

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
