# Peer review of "Evolution of Bacterial Persistence to Antibiotics during a 50,000-Generation Experiment in an Antibiotic-Free Environment"

_antibiotics, 2022, doi:10.3390/antibiotics11040451_

Round 1
Reviewer 1 Report
Dear editor and authors!
The article on title "Evolution of bacterial persistence to antibiotics during a 50,000
generation experiment in an antibiotic-free environment" Is a really interesting and present new look at the antibiotic resistance evolution in a stable environment. I recommend a minor revision.
The manuscript is well written, thought out and laid out. I find the article need a small improves, but after that’s I think it will be to the liking of readers Antibiotics.
I consider following improvements necessary before publication:
General comments (paragraphs):
Title
2: Remove the "dot" from the end of the title.
Abstract
19-20: please specify
30: “about three times the COVID-19 death rate”- remove this sentence- - assumptions supported by assumptions.
36: Is it 10 or 6%? It is assumed that from 10%.
63-67: Recommend remove.
79-86: a paragraph more appropriate to the discussion
Results
- Standardize signatures figures and tables according to antibiotics recommendation.
- Figure 2 could be more readable
Discussion
Discussions can be updated with new research on persistent cells in biofilm (evolution) and strategy of its eradication:
doi: 10.1080 / 22221751.2021.1994355
doi: 10.1007/s00253-020-10349-w
doi: 10.1038/s41591-020-0825-4
doi: 10.1016/j.tim.2021.10.001
Best wishes,
R2
Reviewer 2 Report
Comments and Suggestions for Authors The manuscript describe the evolution of persistence for ampicillin and ciprofloxacin by comparing their effects in the ancestor and evolved clones sampled in each of the twelve populations up to 50,000 generations. Although the work is well performed and easy to follow, additional experiments is necessary. 1.- Since antibiotic behavior in vivo greatly depends on their degree of ionization, lypophylicity, conformational characteristics, concentration, solubility and exposition time. Why the authors comparing two different antibiotics? Why use only one concentration and exposition time? 2.- These variables mentioned could change the conclusions about persistence for ampicillin and ciprofloxacin?Author Response
Please see the attachment.

Round 2
Reviewer 2 Report
The manuscript has been improved. I think that this adds to the body of knowledge and recommend for acceptance.
